# Numerical Study of Wheat Particle Flow Characteristics in a Horizontal Curved Pipe

**Dongming Xu, Yongxiang Li \*, Xuemeng Xu, Yongyu Zhang and Lei Yang**

School of Mechanical and Electrical Engineering, Henan University of Technology, No. 100 Lianhua Street, Zhengzhou 450000, China; 13663842598@163.com (D.X.); xuxuemeng7439@163.com (X.X.); zyongyu@126.com (Y.Z.); leiyang150378@163.com (L.Y.)
**\*** Correspondence: liyongxiang@haut.edu.cn

**Abstract:** Energy consumption is one of the important indicators of green development. The pressure drop and the particle kinetic energy loss in the pipe bend result in high energy consumption of wheat pneumatic conveying. In this paper, the CFD-DEM method is used to study the characteristics of flow field in horizontal pipe bend. The results show that the particles converge together under the force of the curved pipe wall to form a particle rope. With increasing pipe bend ratio $R/D$, the aggregation of particle bundles becomes stronger and the particle spiral phenomenon decreases. The particles impact the pipe wall at an angular position of $\theta = 30$–$60°$ around the bend, and their velocity decreases slowly under the friction resistance of the pipe wall. The velocity loss caused by particles impacting on the pipe wall increases with increasing initial velocity. When the particle mass flow rate is 1.26 kg/s and the gas velocity is 10 m/s, the pressure drop in the bend decreases and then increases with increasing $R/D$. The pressure drop of the bend is smallest for $R/D = 2$ and increases gradually with increasing gas-phase velocity. With increasing of $R/D$, the wall shear force between the particles and the bending pipe decreases and then increases, and the position of the maximum force moves towards the bottom of the bending pipe. The area over which the wall shear force acts continues to decrease because of the aggregation of particle bundles. The research results provide a theory for optimal design and application of pneumatic conveying equipment for wheat particles.

**Keywords:** pneumatic conveying; CFD-DEM; wheat particle; particle flow characteristic; bending pressure drop





## 1. Introduction

Pneumatic conveying is widely used in grain machinery and equipment, such as grain suction machines, grain processing machines, and grain transfer equipment. However, the particle collision, the wall wear, and the pipe pressure drop occur in the pipe bend, which increases energy consumption and reduces grain quality in the system. The flow field characteristics in the pipe bend were studied to reduce the energy consumption of conveying system.

In the past few years, various methods, including DEM and CFD, have been utilized in the study of pneumatic conveying. Compared with traditional experimental methods, CFD methods have many advantages, such as being relatively inexpensive and providing the particle flow status under different operating conditions [1]. The combined DEM-CFD method has been widely used in the study of granular materials' transport characteristics. For example, Li et al. [2] used CFD-DEM to simulate the horizontal flow characteristics of particles in his research and analyzed various factors affecting particle motion. Wei et al. [3] applied the CFD-DEM coupling method to the study of pneumatic conveying of coal particles, and showed that both the pressure variation pattern and the particle beam are affected by the ratio of solid to gas. Kuang et al. [4] utilized a 3D version of a previously proposed 2D model to study the pressure drop and flow patterns in vertical pipelines. Levy

and Mason [5] studied the influence of pipe bending on particle distribution, and they found that particles would accumulate near the outer wall of the pipe downstream of the bend, with the concentration distribution in the pipeline radial being closely related to the pipe bend ratio (the pipe's centerline radius of curvature R divided by its diameter D), the particle size, the pipe layout direction, and other factors; the movement trajectory of the particles after passing through the bend is determined by the particle size, and when there are particles of different sizes in the particle phase, the particles will segregate after passing through the bend. Lee et al. [6] used experiments and two-fluid-model simulations to investigate the flow of polypropylene particles and glass particles in a pneumatic conveying pipe with an L-shaped 90° bend; unlike when passing through a pipe with a smooth bend, the polypropylene particles underwent particle agglomeration, particle agglomeration flow, particle beam formation and particle beam flow successively when passing through the pipe with the L-shaped bend, and the glass particles exhibited completely different flow characteristics. Vashisth et al. [7] used LPT to study the pneumatic conveying characteristics of pulverized coal, glass beads, and polypropylene particles in a 90° pipe bend; the basic characteristics including particle beam, particle segregation, secondary flow, and gas-solid interphase force were obtained by numerical simulation, and the flow patterns of different particles in bend diameters were compared. The results showed that different types of particles show different two-phase flow characteristics under the same working conditions. Zhou et al. [8] used a computational fluid dynamics model (CFD-DEM) to study the effects of rotation strength and particle shape on the erosion caused by a bending pipe. They were able to perform a simulation of a gas–solid flow with the help of vortex strength and the bending direction. Ji and Liu [9] were able to study the effects of different factors, such as the lifting angle, the mass flow rate, and the air velocity, on the particle flow in a pipe with a lifting bend. They found that the pressure drop can gradually increase as the mass flow rate rises. Chu and Yu [10] used a three-dimensional model to analyze computational fluid dynamics to study the flow characteristics of gas phase. Yang and Kuan [11] analyzed the velocity of gas and solid to study the reasons for the generation of particle beams. The power spectrum [12] and wavelet transform [13,14] were used to analyze the gas–solid two-phase flow, revealing the motion characteristics of particle flow. Edward et al. [15] conducted a numerical study of a 50 mm pipe of $R/D = 1.5$ and a longer bend of $R/D = 5$, and the bend pipe was more eroded severely. The mass loss caused by particle erosion in 90° bend pipes is approximately 50 times that of straight pipes [16].

This paper considers the effects of $R/D$, particle phase velocity, and solid mass flow rate on the particle velocity and trajectory, as well as on particle wall shear force and flow–field pressure drop at different angular positions around a pipe bend, which are issues that have rarely been considered until now. These influence laws are helpful not only for understanding the transport mechanism and movement characteristics of wheat particles in bending pipes but also for designing pneumatic conveying systems for wheat particles.

## 2. Experimental Details

### 2.1. Pneumatic Conveying Test Device

Figure 1 is a schematic diagram of the pneumatic conveying device, comprising an air compressor, an air storage tank, an air cooler and dryer, an intake pipe, a vortex flowmeter, a pressure sensor, a silo pump, a pressure transmitter, an acrylic pipe with a bend in it, a control system, a high-speed camera, a cyclone separator, and a receiving bin. Figure 1 reproduced with permission from [17]. The air source was a screw air compressor (BK22-8ZG; Zhejiang Kaishan Compressor Co., Ltd., Quzhou, China) that supplied air with a pressure range of 0–0.8 MPa. First, the air flow entered the silo pump through the upper and lower air intakes, so that the grain particles in the silo pump were fully fluidized. Next, the mixed gas–solid material after fluidization was transported to the conveying pipe. As shown in Figure 2, In the speed measurement experiment, a small amount of

labeled particles were added to the experimental material for easy observation. Using a high-speed camera (dimax HD + PCO, Kelheim, Germany), ultra-clear images were captured at different shooting points on a curved pipe at a camera speed of 1603 fps at a full resolution of 1920 × 1440 pixels. The graphics were processed using Matlab and the average velocity of particles were calculated by the ratio of displacement to time. In the simulation, particle velocity sensors were added to the curved pipe to obtain particle velocities at different times.

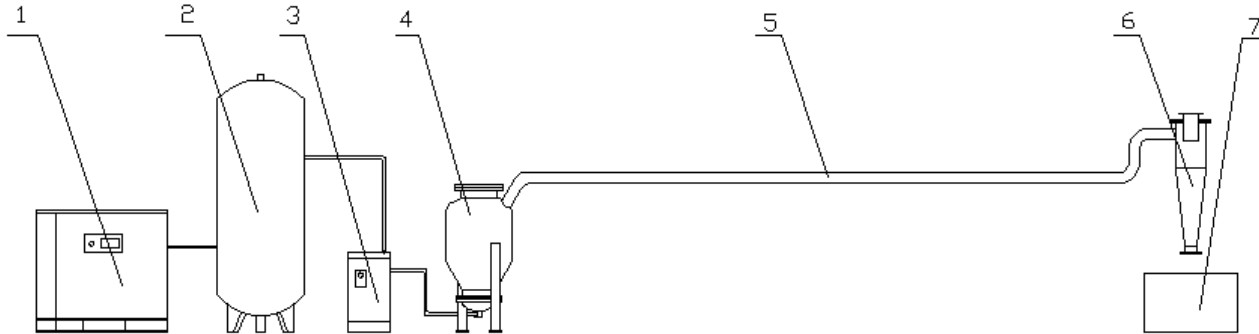

**Figure 1.** Experimental device used for pneumatic conveying: 1—compressor; 2—gas source; 3—Gas cooling dryer; 4—material silo; 5—experimental piping; 6—cyclone separator; 7—receiving bin.

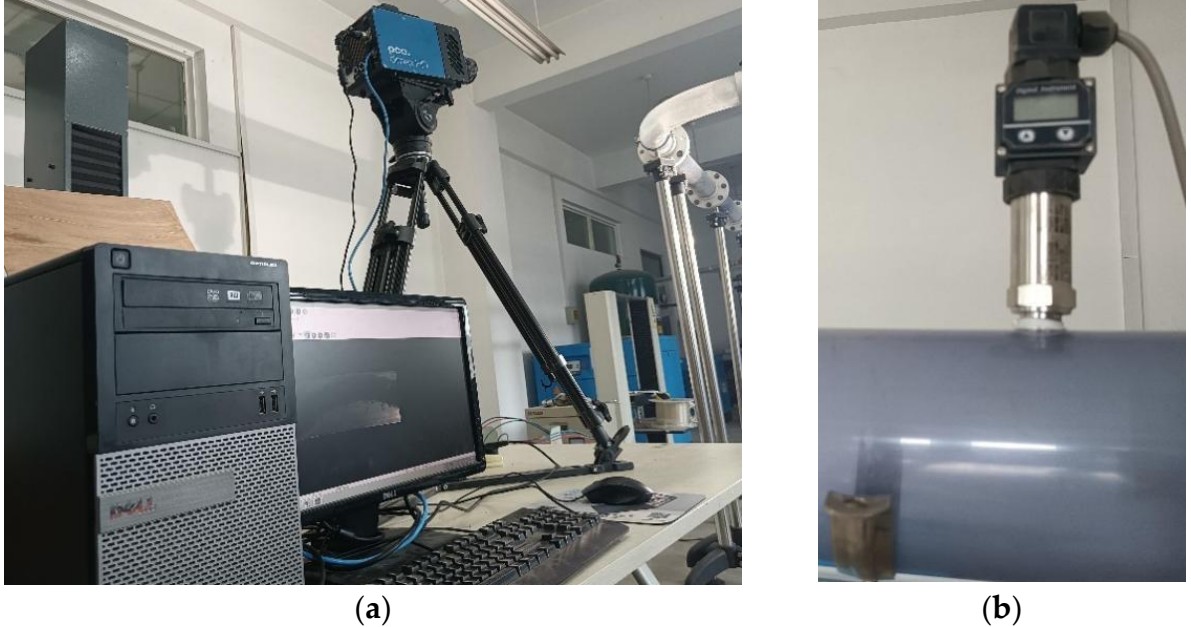

(**a**)     (**b**)

**Figure 2.** Photographs of equipment used in bending test: (**a**) high speed camera; (**b**) pressure sensor.

### 2.2. Material

The chosen experimental material was "Xinmai 26" wheat as studied by the Xinxiang Academy of Agricultural Sciences, Henan Province, China. As shown in Figure 3, the water content of the wheat was measured by a moisture analyzer (QLMD-720A, Remi Scientific, Subang Jaya, Malaysia) and was found to be 10.25%. For the wheat grains, the length of the long was ca 6.2 mm, the length of the short was ca 3.1 mm, and the length of the middle was ca 2.9 mm.

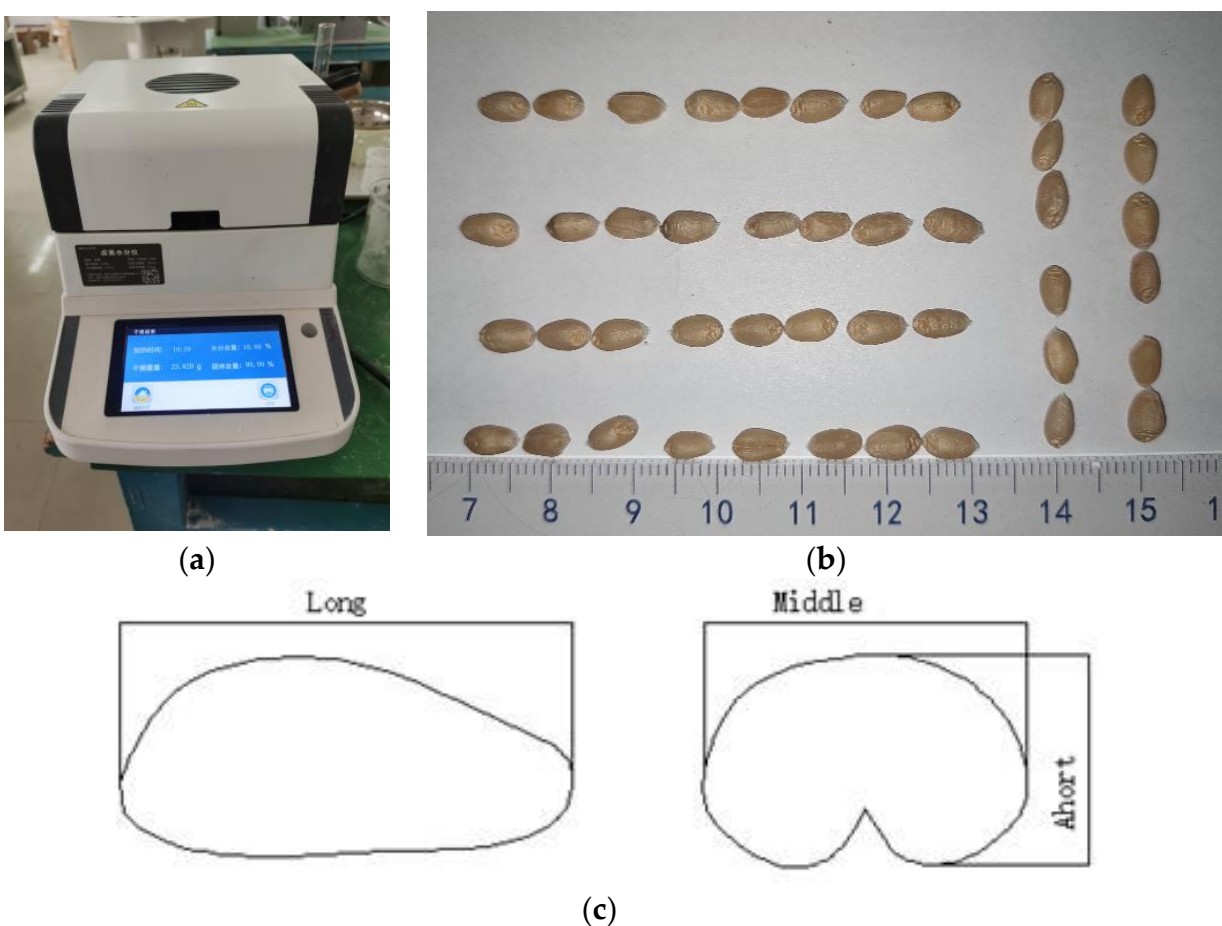

**Figure 3.** (**a**) Wheat moisture measuring instrument and (**b**) physical size measurements (**c**) triaxial dimensions of wheat: Long, meddle and short are the dimensional values of the three directions in the three-dimensional space of the particle.

### 2.3. Test Parameters

The mass flow rates of the particles were set as 0.42 kg/s, 0.84 kg/s, 1.26 kg/s, 1.68 kg/s and 2.1 kg/s, and the particle-phase velocity was as 5 m/s, 10 m/s, 15 m/s, 20 m/s, 25 m/s, and the pipe bend ratio ($R/D$) was set as $R/D$ = 1, 2, 3, 4, 5, and 6.

## 3. Mathematical Model

### 3.1. Coupled CFD-DEM Method

Numerical methods applied to the velocity, trajectory, and distribution of particles in horizontal bends, as well as the particle–wall force and flow-field pressure drop. In the study, the gas phase was controlled using Navier–Stokes equations, while the solid phase was controlled using Newton's second law.

### 3.1.1. Particle-Phase Control Equations

Particle-phase were considered as discrete phases, and the equation governing their motion per unit mass are based on Newton's second law [18], i.e.,

$$m\frac{du_p}{dt} = F_D + F_P + F_{VM} + F_G + G_P \tag{1}$$

$$I_p\frac{d\omega_p}{dt} = T_p \tag{2}$$

where $F_D$, $F_{VM}$, $G_p$, $F_P$, $F_G$, $T_p$ are respectively the drag force, added mass force, buoyancy force, pressure gradient force, force of gravity, and the torque.

Drag is the main force that a gas exerts on a solid [19], and it is given by

$$F_D = \frac{18\mu}{\rho_p d_p^2} \frac{C_d Re_p}{24} \left(u_g - u_p\right) \tag{3}$$

where $u_p$, $d_p$, $\rho_p$ are the velocity, the equivalent diameter the density of the wheat grains, and $Re_p$ is the solid-particle Reynolds number given by

$$Re_p = \frac{\rho_g d_p |u_p - u_g|}{\mu} \tag{4}$$

With the varies of the Reynolds number [19,20], The drag coefficient $C_d$ is given by

$$C_d = \begin{cases} \frac{24}{Re_p}\left[1 + 0.15 Re_p^{0.687}\right] & Re_p < 1000 \\ 0.44 & Re_p \geq 1000 \end{cases} \tag{5}$$

Pressure gradient force on particles caused by pressure gradient in the flow field is given by

$$F_P = \left(\frac{\rho_g}{\rho_p}\right) \nabla p \tag{6}$$

and the added mass force on a particle is given by

$$F_{VM} = \frac{1}{2} \frac{\rho_g d \left(u_g - u_p\right)}{\rho_p dt} \tag{7}$$

However, compared to air, particle density is higher, so and the pressure gradient force, added mass force, and buoyancy force can be ignored in the simulation process.

Finally, the force of gravity acting on a particle is given by

$$G_p = \frac{\pi d_p^3 \rho_p g}{6} \tag{8}$$

3.1.2. Gas-Phase Control Equation

The gas phase is considered as a continuous phase and solved based on the Navier Stokes equation [21].

The mass conservation equation is

$$\frac{\partial}{\partial t}\left(\varepsilon_g \rho_g\right) + \nabla \cdot \left(\varepsilon_g \rho_g u_g\right) = 0 \tag{9}$$

and the momentum conservation equation is

$$\frac{\partial}{\partial t}\left(\varepsilon_g \rho_g u_g\right) + \nabla \cdot \left(\varepsilon_g \rho_g u_g u_g\right) = -\varepsilon_g \nabla p + \nabla \cdot \left(\varepsilon_g \tau_g\right) + \varepsilon_g \rho_g g - S \tag{10}$$

where, $\rho_g$, $p$, $\tau_g$ and $u_g$ are the density, pressure, viscosity, and velocity of the gas respectively, $g$ is the gravity, and $S$ is the momentum exchange between gas and particles.

The continuous-phase voidage $\varepsilon_g$ is given by

$$\varepsilon_g = 1 - \sum_{k=1}^{n} V_{p,k} / V \tag{11}$$

where, $V_{p,k}$ is the volume of particle $k$ in a control unit, $V$ is the volume of the control unit, and $n$ is the number of particles therein.

The momentum exchange between gas and particles is given by

$$S = \frac{1}{\Delta V} \sum_{i=1}^{n} \left(\vec{F}_D + \vec{F}_G\right) \tag{12}$$

where, $n$ is the number of particles, and $\Delta V$ is the volume of a control unit therein.

3.1.3. CFD-DEM Contact Force Model

In discrete element simulation, two simplified models and equations are commonly used to describe the contact force: the linear elastic damper model and the nonlinear Hertz–Mindlin–Deresiewicz model [22–24]. Figure 4 shows a schematic diagram of the contact between particles $p$ and $q$.

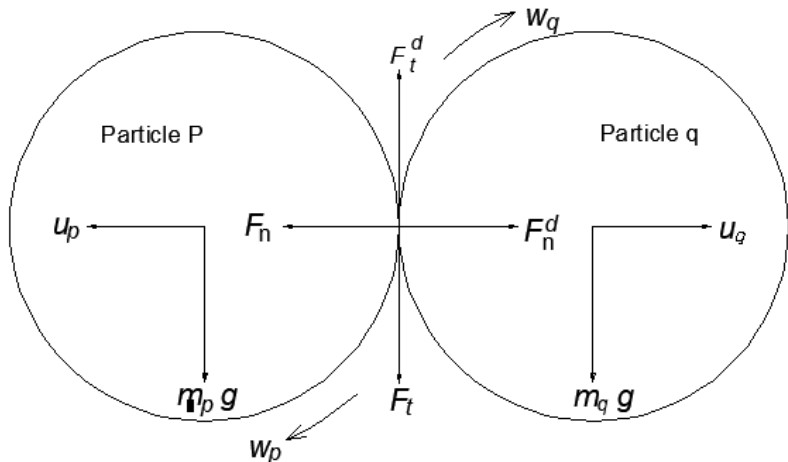

**Figure 4.** Schematic of the contact between particles $p$ and $q$.

The total particle contact force is given by

$$F_{c,q}^{p} = F_n + F_n^d + F_t + F_t^d \tag{13}$$

First, the normal force $F_n$ is represented as follows

$$F_n = \frac{4}{3}E^*\sqrt{R^*}\delta_n^{2/3} \tag{14}$$

$$\delta_n = R_p + R_q - \left|\vec{r}_p - \vec{r}_q\right| \tag{15}$$

$$E^* = \left[\frac{1 - v_p^2}{E_p} + \frac{1 - v_q^2}{E_q}\right]^{-1} \tag{16}$$

$$R^* = \left[\frac{2}{R_p} + \frac{2}{R_q}\right]^{-1} \tag{17}$$

where, $F_n$ is the normal contact force, $\delta_n$ is the amount of overlap between particles, $R_p$ and $R_q$ are the particle radii, $\vec{r}_p$ and $\vec{r}_q$ are the vectors of particles $p$ and $q$, $E_p$ and $E_q$ are the Young's modulus of the two particles, $E^*$ is the equivalent Young's modulus, $v_p$ and $v_p$ are the Poisson's ratio of the two particles.

Next, the normal force $F_n^d$ is represented as follows

$$F_n^d = -2\sqrt{\frac{5}{6}}\frac{\ln e}{\sqrt{(\ln e)^2 + \pi^2}}\sqrt{s_n m^*}v_{npq} \tag{18}$$

$$m^* = \left(\frac{1}{m_p} + \frac{1}{m_q}\right)^{-1} \tag{19}$$

$$s_n = 2E^*\sqrt{R^*\delta_n} \tag{20}$$

where, $e$ is rebound coefficient, $m^*$ is the equivalent mass of particles, $p$ and $q$, $s_n$ is the constant stiffness, $v_{npq}$ is the normal component of velocity.

Next, the tangential force $F_t$ is given by

$$F_t = \begin{cases} -\delta_t S_t & |F_t| < \mu_s |F_n| \\ \mu_s |F_n| \frac{v_{tpq}}{|v_{tpq}|} & |F_t| \geq \mu_s |F_n| \end{cases} \tag{21}$$

$$S_t = 8G^* \sqrt{R^* \delta_n} \tag{22}$$

$$G^* = \frac{2 - v_p^2}{G_p} + \frac{2 - v_q^2}{G_q} \tag{23}$$

where, $S_t$ is the tangential stiffness, $G_p$ and $G_q$ are the shear moduli of the two particles, $G^*$ is the equivalent shear modulus, $\delta_t$ and $\mu_s$ are the tangential overlap and the sliding friction coefficient of the two particles, $v_{tpq}$ is the tangential velocity of of the two particles.

Finally, tangential damping force $F_t^d$ is given by

$$F_t^d = -2\sqrt{\frac{5}{6}} \frac{\ln e}{\sqrt{(\ln e)^2 + \pi^2}} \sqrt{s_t m^*} v_{tpq} \tag{24}$$

### 3.1.4. Particle–Wall Contact Model

As shown in Figure 5, when a wheat particle contacts the pipe, the angles of particle velocity direction and tangential direction of the contact point are $\alpha_1$ and $\alpha_2$. In CFD simulation, the collision recovery coefficient between particles and pipe walls is usually used to solve for the particle movement. The particle velocity changes after the particle collides with the pipe wall.

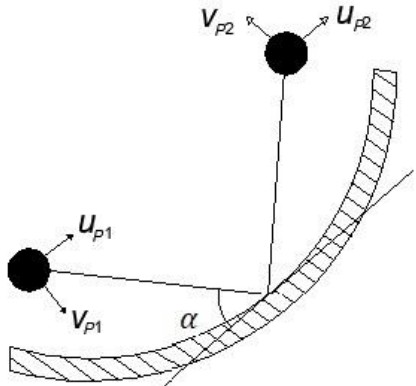

**Figure 5.** Collision process between a grain and the pipe wall.

In the collision, to measure the particle kinetic energy loss, the collision recovery coefficients can be used, i.e.,

$$e_n = \frac{u_{p2}}{u_{p1}} \tag{25}$$

$$e_t = \frac{v_{p2}}{v_{p1}} \tag{26}$$

The particle impact–rebound coefficients [25] are given by

$$e_n = 0.993 - 0.0307\alpha + 4.75 \times 10^{-4}\alpha^2 - 2.61 \times 10^{-6}\alpha^3 \tag{27}$$

$$e_t = 0.988 - 0.0290\alpha + 6.43 \times 10^{-4}\alpha^2 - 3.56 \times 10^{-6}\alpha^3 \tag{28}$$

where, $e_n$ and $e_t$ is the recovery coefficient of normal and tangential, $\alpha$ is the particle impact angle.

### 3.2. Calculation Conditions

#### 3.2.1. Geometric Model and Meshing

Figure 6 shows the geometrical model of a 90° standard bend. The geometrical model and grid division were constructed according to the real size of the pipeline in the test. The bending radius of the pipe is expressed as *R*, and the pipe diameter is *D* = 100 mm. To verify the accuracy of the experimental results with grid size, the calculation is stable when the grid size is larger than the particle size. When the grid size is smaller than the particle size, instability occurs in the calculation. Therefore, to maintain stability in the calculation, the fluid mesh volume should be greater than the particle volume. The EDEM wheat grain model is constructed based on the true particle size. Meanwhile, to obtain the contact characteristics between particles and the pipe wall, the ANSYS ICEM is used to partition and refine only the hexahedral mesh in the near wall region, as shown in Figure 7 [9,26,27].

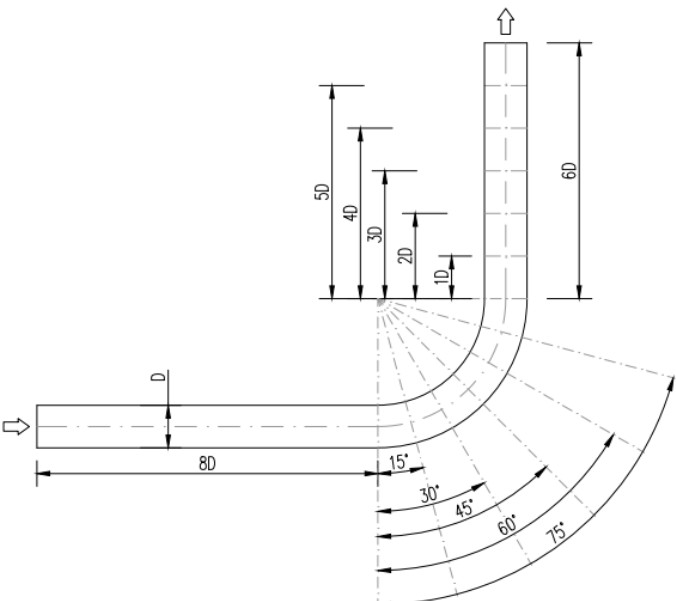

**Figure 6.** Geometrical model of pipe bend.

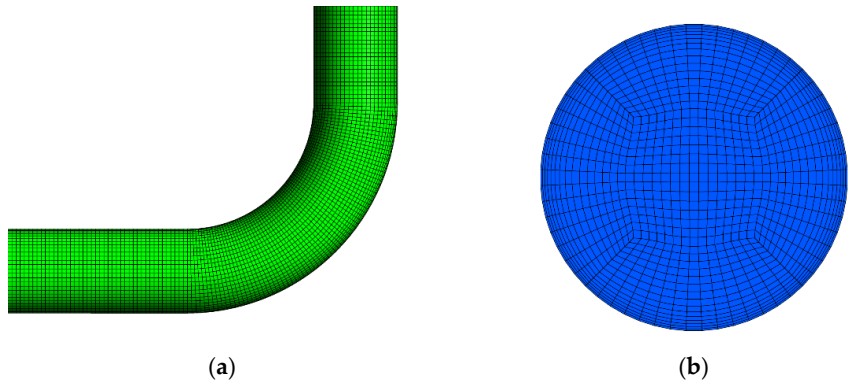

(**a**)                                                          (**b**)

**Figure 7.** Numerical grid of (**a**) longitudinal section and (**b**) transverse section.

Considering the irregularity of wheat particle shape and size [28], non-spherical particles were used to construct the grain model. To reduce calculation time and consider the authenticity of simulation, the EDEM model of a wheat particle was constructed with nine spheres according to the spherical superposition method, as shown in Figure 8.

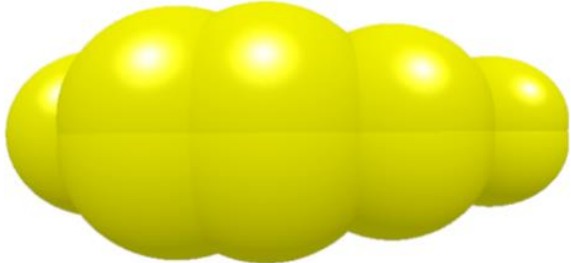

**Figure 8.** Wheat particles in EDEM.

### 3.2.2. Parameter Calibration

In the preliminary research, our team calibrated the parameters of wheat grains and conducted corresponding simulation studies. EDEM particle contact parameter calibration program: The static angle of rest of wheat grains was used as the response index. The required discrete element parameters for the DEM model were determined and Plackett-Burman design was used to test the significance of the parameters to be calibrated. The Steepest Ascent test was used to narrow down the optimization range of parameters. The second-order multiple regression model was established to test the results of the Central composite design, and the DEM parameters were optimized for the regression model, as shown in Figure 9.

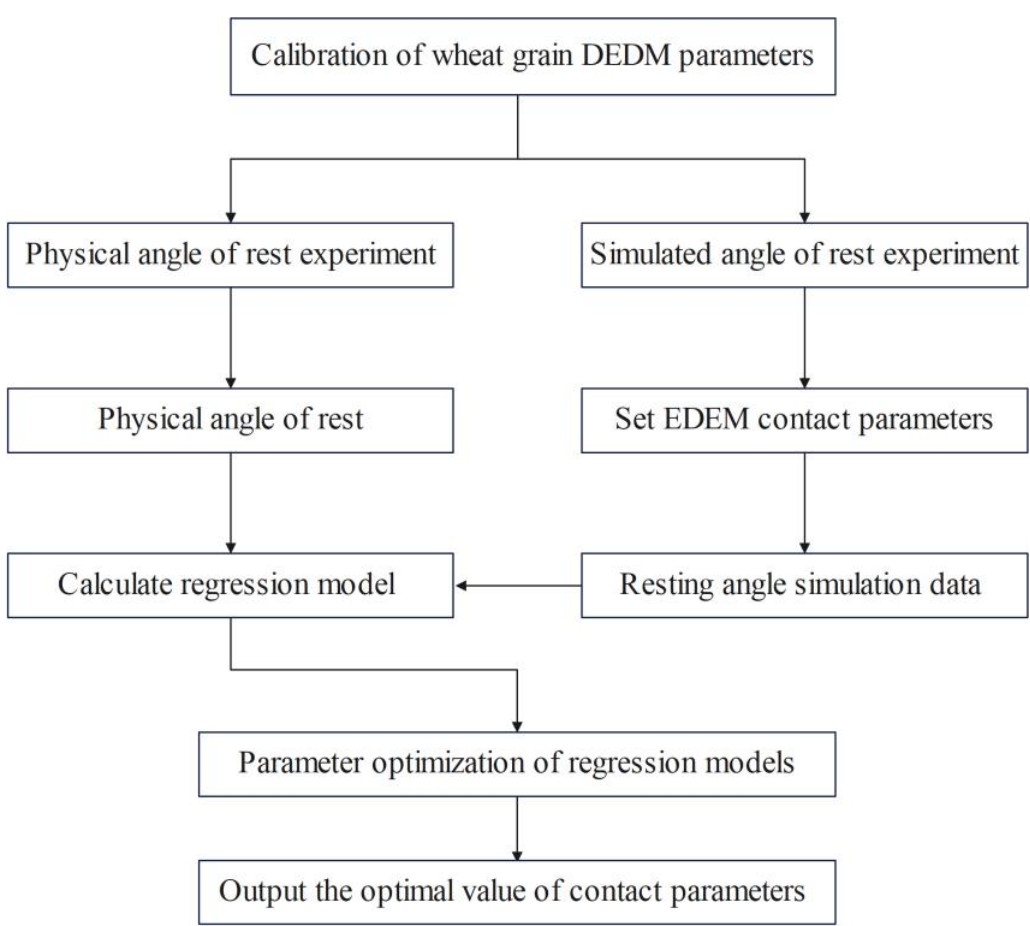

**Figure 9.** DEM particle calibration program.

### 3.2.3. Number of Grids and Grid Independence

We conducted grid independent research. To achieve this, we created three grid models with different cell numbers for the $R/D = 2$ bend, including 178,521, 273,129, and

370,124 cells, as shown in Table 1. Particle inlet speed was set at 10 m/s. The particles passing through the bend was simulated. The average velocity of particles at different bending angles of 0°, 15°, 30°, 45°, 60°, 75°, and 90° was measured to verify the influence of grid number on the simulation results. The results were shown in Figure 10. There was no significant difference in particle velocities, and the results were independent of the grid. After proper consideration, the number of grids with $R/D$ = 1, 2, 3, 4, 5, 6 was determined to be 251,301, 273,129, 291,109, 320,257, 342,468, 367,110, respectively.

**Table 1.** Mesh independence study.

| Total Number of Grids | Total Number of Nodes |
| --- | --- |
| 178,421 | 184,359 |
| 273,129 | 284,014 |
| 370,124 | 385,011 |

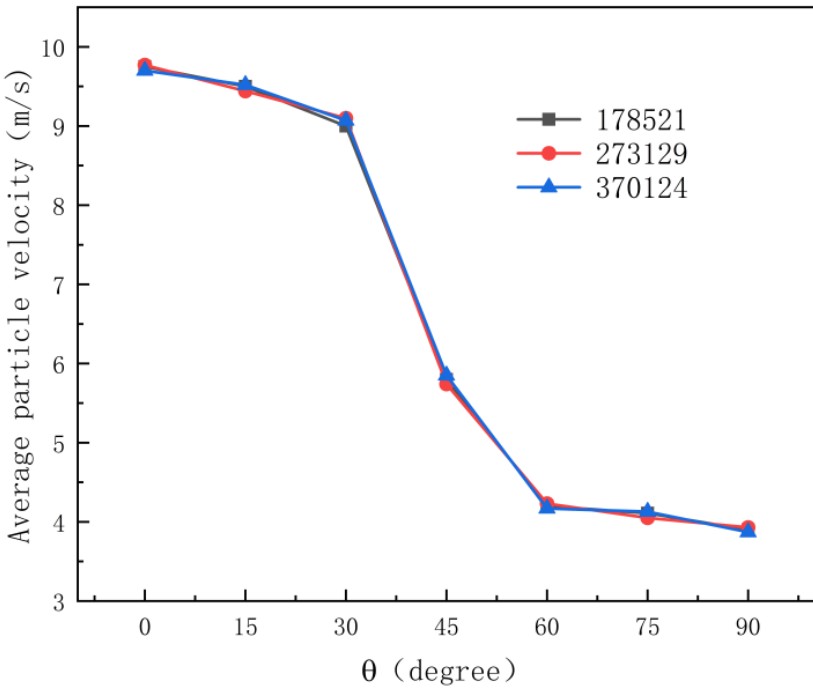

**Figure 10.** Comparison of particle velocities along curved pipes under different grid numbers.

3.2.4. Simulation Setup and Test Design

Based on the pressure solver and finite volume method, this article uses an algorithm with pressure velocity coupling to solve the control equation. The time step in CFD is an important parameter in transient simulation. If the time step is too large, it cannot solve the transient changes. A smaller time step can maintain the stability of the solver and obtain the true solution. The Courant number is commonly used to estimate the time step size. The time step of EDEM cannot be greater than the time step of Fluent. The simulation step size of this article is reflected in Table 2. The particle factory type was set to Unlimited Number. Numerical simulation parameters are given in Table 2 [17].

**Table 2.** Simulation parameters.

| | Project | Details | Indicators | Numerical Value |
|---|---|---|---|---|
| CFD | Materials | Air | Density [kg/m$^3$] | 1.225 |
| | | | Viscosity [kg/m$^2$·s] | $1.7894 \times 10^{-5}$ |
| | | Solid | Density [kg/m$^3$] | 1200 |
| | Boundary conditions | Velocity inlet | Velocity magnitude [m/s] | 5~25 |
| | | Turbulence | Turbulence intensity | 4% |
| | | | Hydraulic diameter [mm] | 100 |
| | | Pressure outlet | Pressure | 0 |
| | | Wall | Wall movement | Still wall |
| | | | Shear condition | No slip |
| | | | Roughness [mm] | 0.0015 |
| | | | Roughness constant | 0.5 |
| | | Time step | Fixed time step [s] | $1 \times 10^{-4}$ |
| DEM | Materials | Wheat particle | Poisson's ratio | 0.29 |
| | | | Shear modulus [Pa] | $5 \times 10^8$ |
| | | | Density [kg/m$^3$] | 1350 |
| | | Walls | Poisson's ratio | 0.3 |
| | | | Shear modulus [Pa] | $7 \times 10^{10}$ |
| | | | Density [kg/m$^3$] | 1200 |
| | Contact | Particle–particle | Recovery coefficient | 0.5 |
| | | | Static friction coefficient | 0.35 |
| | | | Rolling friction coefficient | 0.05 |
| | | | Interactive contact model | Hertz–Mindlin (no slip) |
| | | Particle–wall | Coefficient of recovery | 0.50 |
| | | | Static friction coefficient | 0.30 |
| | | | Rolling friction coefficient | 0.05 |
| | | | Contact model | Hertz–Mindlin (no slip) |
| | Particle Factory | Parameter setting | Type of factory | Dynamic/unlimited number |
| | | | Mass flow [kg/s] | 0.42~2.1 |
| | | | Fixed time step [s] | $5 \times 10^{-6}$ |
| | | | Mesh size [mm] | 10 |
| | | | Particle equivalent diameter [mm] | 4 |

### 3.2.5. Computational Procedure

In the CFD-DEM coupling process of this article, FLUENT software (https://www.ansys.com/products/fluids/ansys-fluent) calculates the gas velocity and flow field pressure based on the initial conditions of the gas flow field in Table 1. The calculation results are transmitted to EDEM software (https://altair.com/edem-applications) to solve the particle velocity and position information, and the particle information is transmitted back to FLUENT software. The software calculates the void fraction and gas–solid interaction force of each grid at this time step based on the gas flow field and particle information, completing the momentum exchange between the gas and solid phases. The gas-solid interaction force data are transferred to EDEM software to solve the particle information for the next time step; at the same time, the porosity and momentum exchange is transferred to FLUENT to calculate the gas flow field for the next time step, as shown in Figure 11.

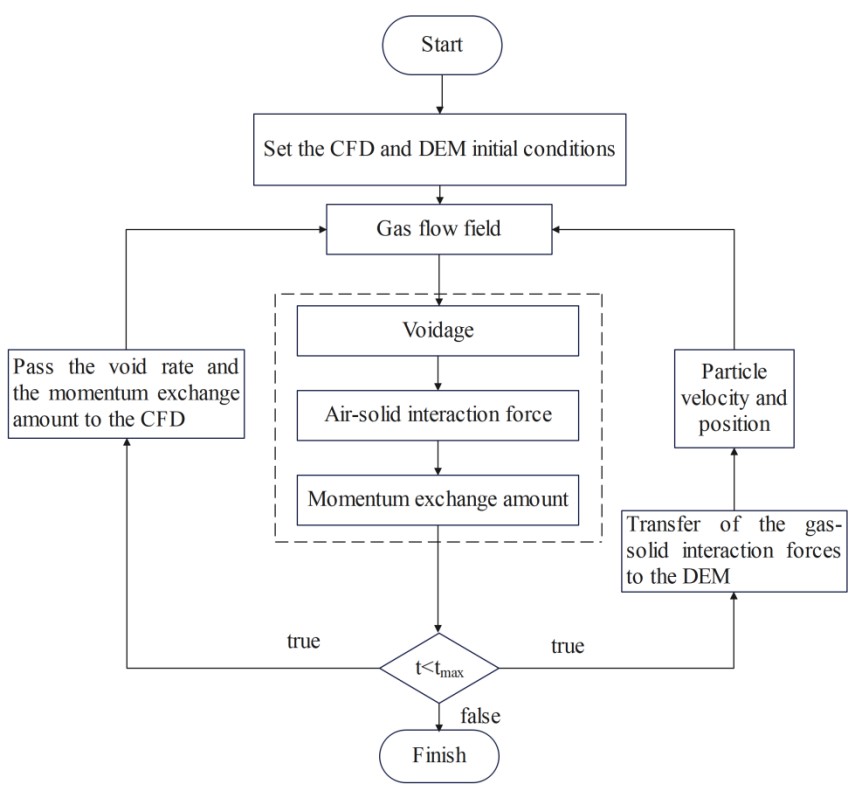

**Figure 11.** Computational procedure.

## 4. Results and Discussion

### 4.1. Model Validation

The feasibility of using the CFD-DEM model to study wheat granules was analyzed and the feasibility of the model from particle velocity and particle trajectory were verified. Figure 12 shows the simulation and experimental views of the particle flow trajectory for a mass flow rate of 0.84 kg/s, a particle-phase velocity of 10 m/s, and $R/D = 2$. These results show that under the same conditions, the simulated particle flow trajectory in the bend is similar to the experimental trajectory, and so the numerical simulation method can be used to study the particle flow trajectory in the bend.

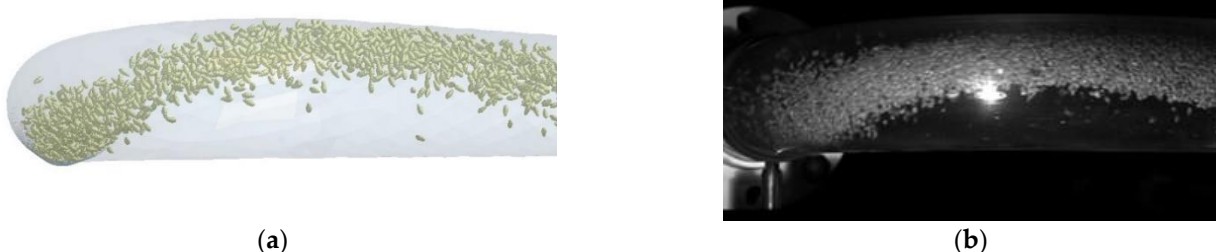

(**a**)            (**b**)

**Figure 12.** (**a**) Simulation and (**b**) experimental data views of particle flow trajectory in bend.

Figure 13 shows experimental and simulation results for the particle velocity for a mass flow rate of 1.26 kg/s, $R/D = 4$, and a particle-phase velocity of 10 m/s. These results show that under the same conditions, the simulated-average particle velocity at different values of the angle $\theta$ around the pipe bend is similar to the experimental data, and so the numerical simulation method can be used to study the particle velocity in the bend.

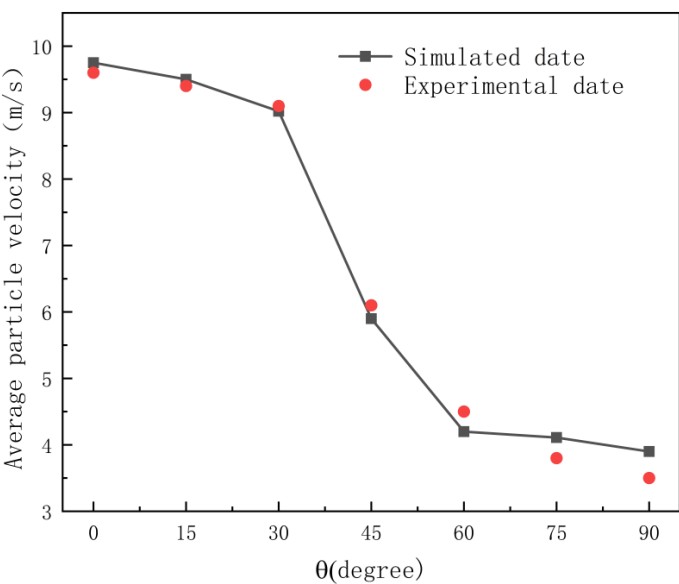

**Figure 13.** Comparison of simulated and experimental data.

*4.2. Influence of Pipe Bend Ratio on Particle Trajectory*

Figure 14 shows the radial distribution of particles in the 90° horizontal bend and downstream. Under the action of gravity, particles at $\theta = 0–15°$ move along the lower part of the bend. At $\theta = 30°$, the particles contact the pipe wall and form a particle bundle. The particles can move through the pipe's upper part and then spiral along the inner wall.

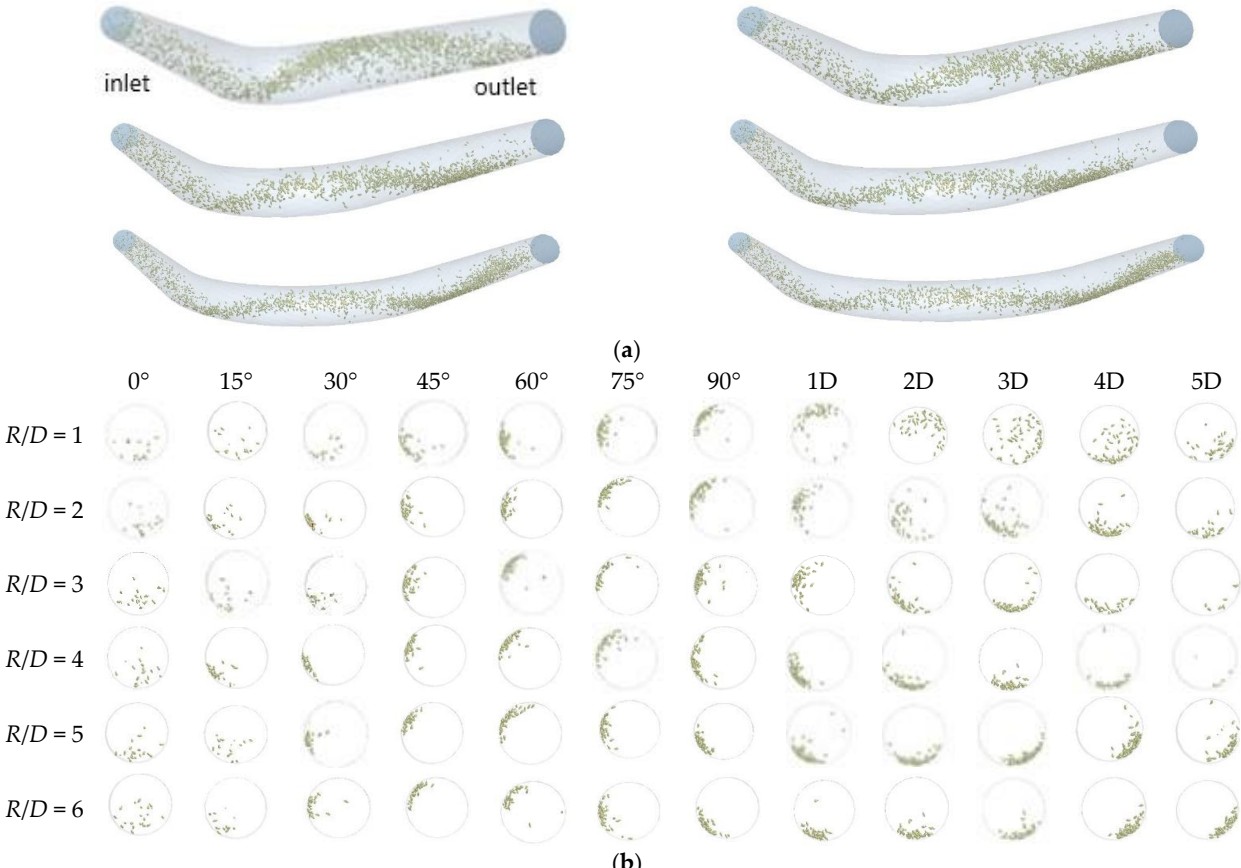

**Figure 14.** (**a**) Particle trajectory and (**b**) radial distribution of particles at different values of angle $\theta$ around pipe bend.

The spiral starting position of particles varies with the variation of $R/D$. As shown in Figure 14b, for $R/D = 1$, the particles move upward at $\theta = 45°$, and this value decreases gradually with the increasing of $R/D$. For $R/D = 1$, the particles reach the top of the bend along the wall of the bend and form a complete spiral. For $R/D > 1$, the particles reach the top of the bend along the wall and then move down the outer wall of the bend. The larger the value of $R/D$, the greater the influence of the pipe bend on particle aggregation and the stronger the particle bundle. Compared with the experimental results of particle trajectories in curved pipes [29], the simulation results of particle motion trajectories are consistent with the experimental results.

*4.3. Influence of Pipe Bend Ratio on Particle Velocity*

4.3.1. Particle Velocity at Different Values of Angle around Pipe Bend

As shown in Figure 15, the particles have the same velocity at the start of the bend ($\theta = 0°$). When they move to $\theta = 30–60°$, the particles impact the pipe wall, their movement direction changes. The particle velocity drops rapidly. At $\theta = 60–90°$, the particles are affected by the friction resistance of the pipe wall. The particle velocity decreases slowly. Compared with the experimental results of particle velocity in the bend [30], the simulation results of particle motion in the pipeline are consistent with the experimental results.

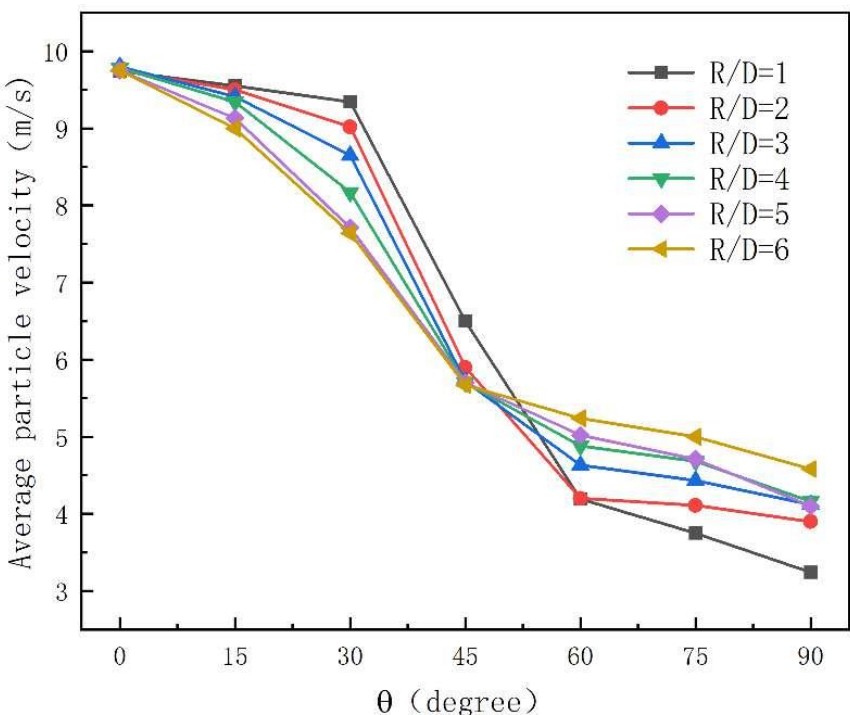

**Figure 15.** Particle velocity at different values of angle $\theta$ around pipe bend.

4.3.2. Effect of Initial Particle Velocity on Particle Velocity Loss

Figure 16 shows changes in axial velocity of particles with different initial speeds of 5 m/s, 10 m/s, 15 m/s, 20 m/s, and 25 m/s for $R/D = 2$. As can be seen, the higher the particle initial velocity, the more it decreases in the bend. At $\theta = 30–60°$, the particle velocity decreases for impacting on the pipe wall, and because the frictional resistance of the bending pipe is proportional to the square of the velocity, greater particle velocity means greater frictional resistance and hence greater loss of particle velocity.

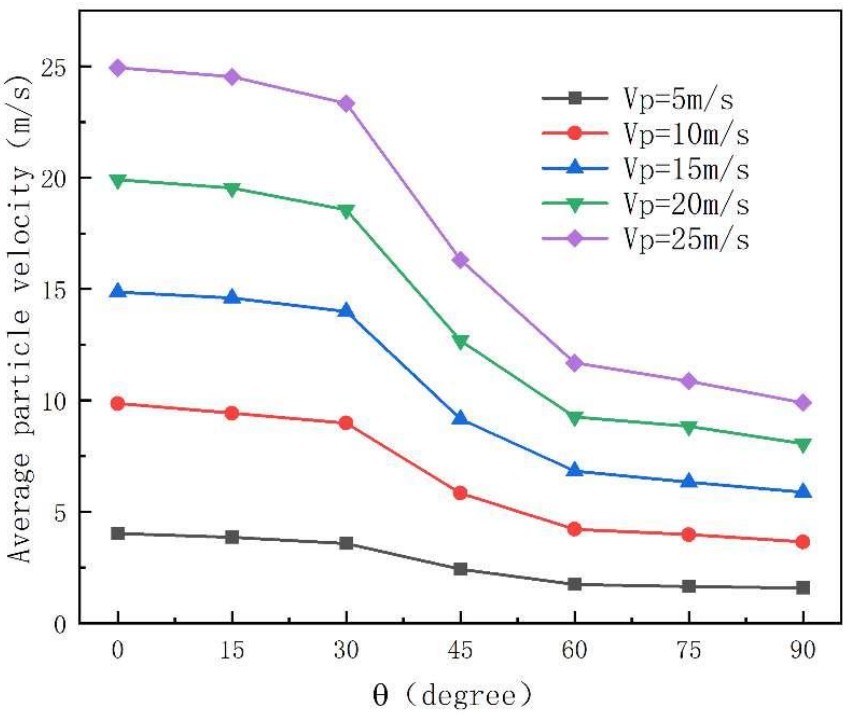

**Figure 16.** Variation of particle velocity in axial direction for $R/D$ = 2.

*4.4. Elbow Pressure Drop*

4.4.1. Effect of $R/D$ on Pressure Drop in Pipe

As shown in Figure 17, for $R/D$ = 1, 2, and 3, there is a significant pressure difference between the inner and outer walls of the pipe bend. The radial pressure gradient is obvious and the turbulent kinetic energy is large. For $R/D$ = 4, 5, and 6, the radial pressure distribution becomes gradually uniform. When the particle mass flow rate is 1.26 kg/s and the gas-phase velocity is 10 m/s, the pressure drop in the bend decreases and then increases with increasing $R/D$. The pressure drop in the bend is smallest for $R/D$ = 2, because (i) for $R/D$ = 1, the pipe bend exerts greater resistance om the fluid, and (ii) for $R/D$ > 2, the particles have farther to travel and so the fluid pressure is transformed more into particle dynamics.

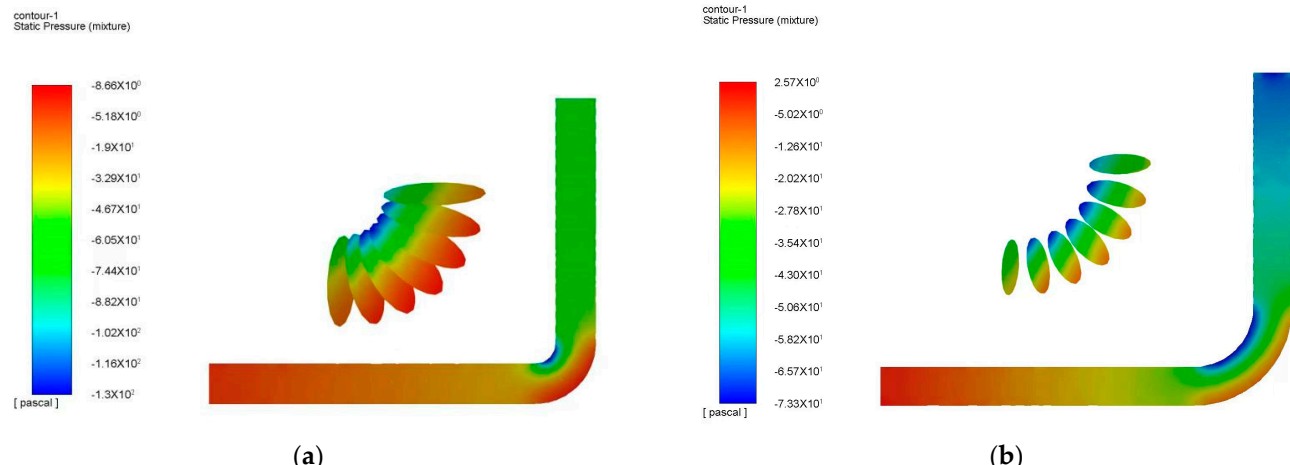

(**a**)  (**b**)

**Figure 17.** *Cont.*

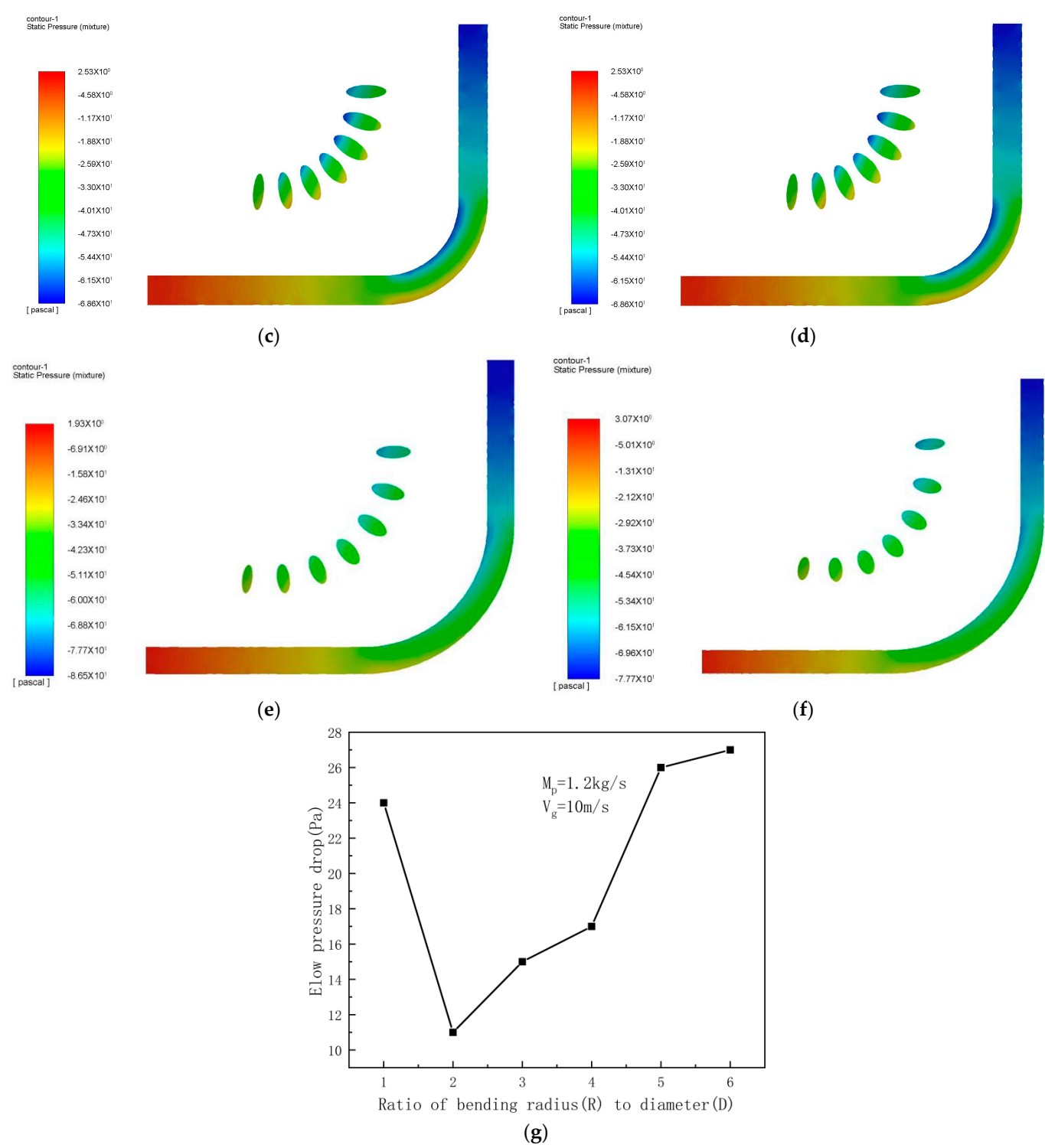

**Figure 17.** Flow-field pressure distribution and pressure drop around horizontal pipe bend for bend ratio $R/D$ = (**a**) 1, (**b**) 2, (**c**) 3, (**d**) 4, (**e**) 5, (**f**) 6; (**g**) variation of pressure drop in pipe bend with $R/D$.

### 4.4.2. Effect of Gas-Phase Velocity on Pressure Drop in Pipe

Figure 18 shows the pressure drop with the variation of gas-phase velocity. When Mp = 1.26 kg/s and $R/D$ = 2, the pressure drop of the bend increases gradually as the gas phase velocity rises. Compared with the pressure drop reported in the experiment [31], the calculated results show a consistent trend with the experimental results.

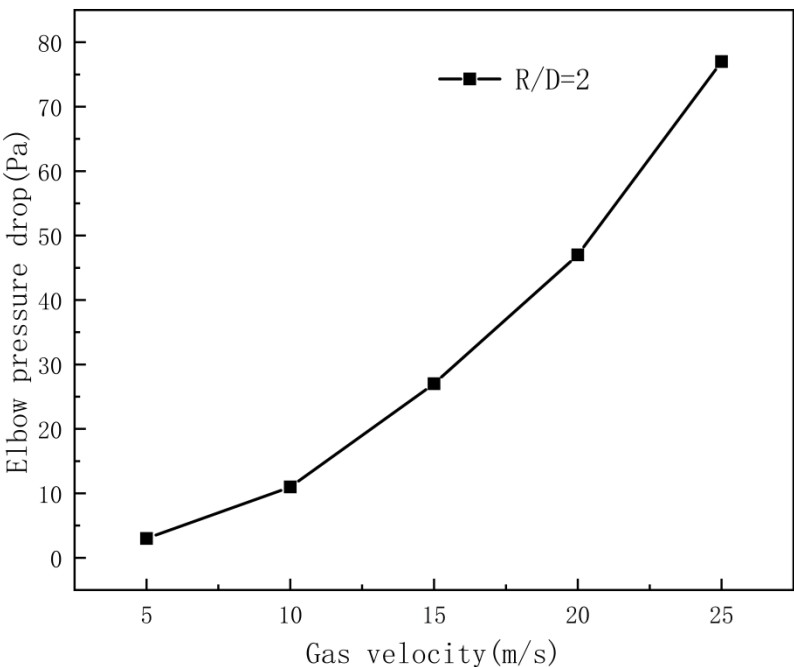

**Figure 18.** The pressure drop with the variation of gas-phase velocity.

### 4.4.3. Pressure Drop of the Pipe Bend under Different Particles Mass Flow Rates

The mass flow rates of particles were shown in Figure 19. The horizontal pipe bend's pressure drop was also affected by the different values of the mass flow rates. When the latter increases in the range of 0.42~2.1 kg/s, the pressure drop in the horizontal bend decreases and then increases. The reason is that when the particle mass flow rate is low, all the particles have a fierce collision with the pipe wall. The pressure drop of the elbow is large. As the mass flow rate increases, more pressure is required to complete the transportation of particles. When the mass flow rate is 1.26 kg/s, the pressure drop of the pipe bend is the smallest. For a given gas-phase velocity, increasing the particle mass flow rate leads to particle deposition and increases the pressure drop in the elbow. Compared with the experimental results of unit pressure drop in bent pipes [24,32], the simulation results are consistent with the experimental results when the mass flow rate is large.

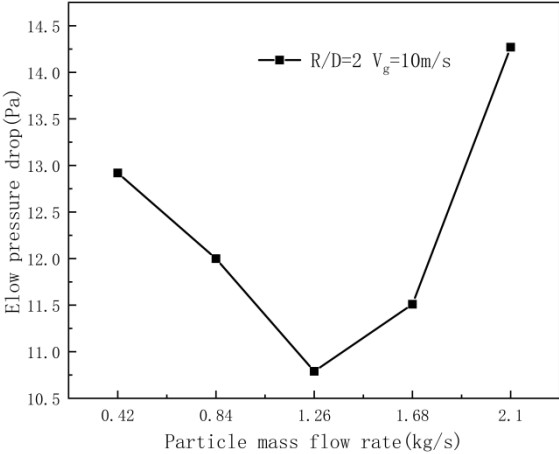

**Figure 19.** Variation of pressure drop in pipe bend with mass flow rates.

### 4.5. Shear Force between Particles and Pipe Wall

Figure 20 shows the shear force nephogram of the shear force between the particles and the pipe wall under different values of $R/D$. With increasing $R/D$, the wall shear force

decreases and then increases, and the distribution of shear forces varies with the variation of $R/D$. The area over which the wall shear force acts keeps decreasing because of the aggregation of particle bundles.

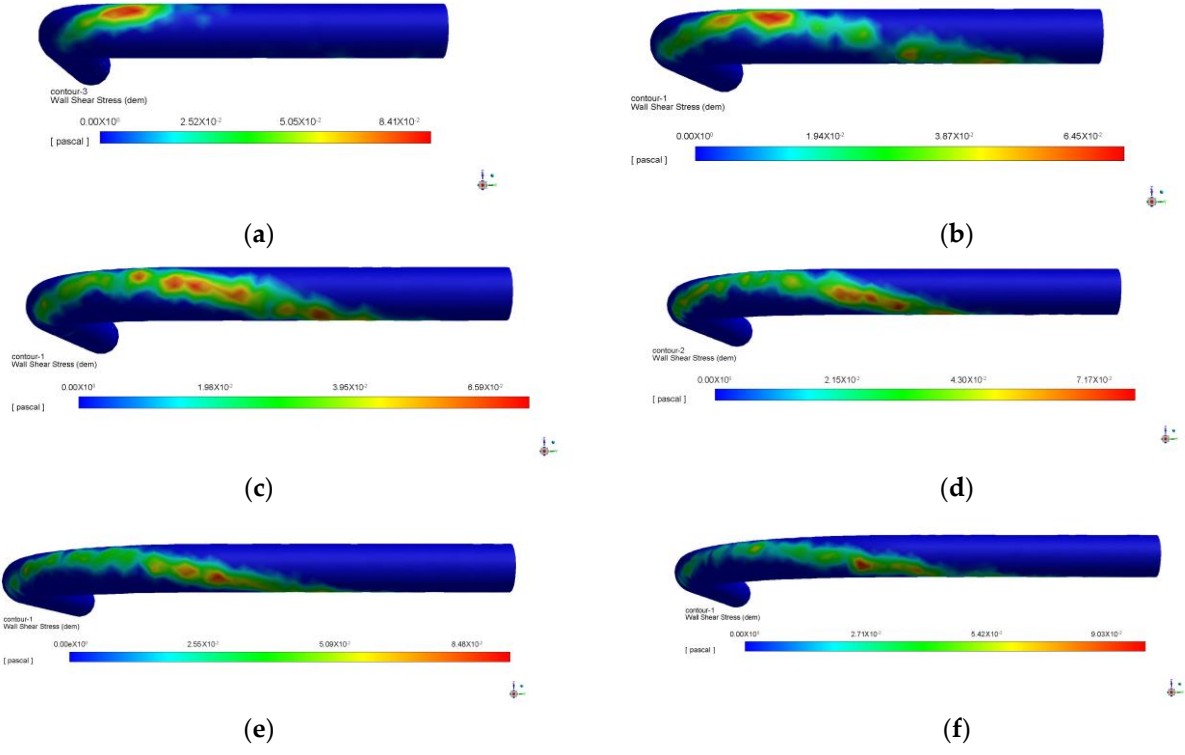

**Figure 20.** Distribution of shear force between particles and pipe wall under different values of pipe bend ratio: bending diameter ratios: $R/D$ = (**a**) 1, (**b**) 2, (**c**) 3, (**d**) 4, (**e**) 5, and (**f**) 6.

## 5. Conclusions

In this study herein, the CFD-DEM was used to study wheat-particle velocity, trajectory, and radial distribution, pipe bend pressure drop, and particle wall–shear force in the pipe bend. From the reported results and discussions, the following conclusions are drawn.

When the pipe bend ratio $R/D$ is increased from 1 to 6, the starting position of the particle spiral is the same, but particles are more aggregated and less dispersed. With increasing $R/D$, (i) the initial upward motion angle of particles decreases gradually, (ii) the upward component of particle velocity decreases, (iii) the particle spiral effect is weakened through the horizontal bend, and (iv) the particle beam is enhanced.

When the particles reach $\theta$ = 30–60° around the bend, they impact the pipe wall and their velocity drops rapidly. At $\theta$ = 60–90°, the particle velocity decreases due to the frictional resistance of the pipe wall, and their velocity decreases slowly. The greater the collision velocity between particles and the pipe wall, the greater the loss of particle kinetic energy, and so the greater the particle velocity, the greater the friction resistance and hence the greater the loss of particle velocity.

When the particle mass flow rate is 1.26 kg/s and the velocity of gas phase is 10 m/s, the pressure drop in the bend decreases and then increases with increasing $R/D$. The pressure drop in the bend is smallest for $R/D$ = 2 and increases gradually with increasing gas-phase velocity.

With increasing $R/D$, the wall shear force between the particles and the bending pipe decreases and then increases, and the position of its maximum value moves toward the bottom of the bending pipe. The area over which the force between the particles and the pipe wall moves towards the bottom of the bending pipe. The area of the wall shear force acts keeps decreasing because of the aggregation of particle bundles.

**Author Contributions:** D.X.: Data curation, Writing-original draft preparation, Software, Y.L.: conceptualization, Methodology, Supervision. X.X.: Investigation, Methodology, Y.Z.: conceptualization, guidance. L.Y.: Investigation, Methodology. All authors have read and agreed to the published version of the manuscript.

**Funding:** This research was funded by the financial support projects which comes from the National Key Research and Development Program of China (grant no. 2018YFD0400704 and grant no. 2022YFD2100201).

**Data Availability Statement:** The data collected and used to support the research results can be provided at the request of the corresponding author.

**Conflicts of Interest:** The authors declare that they have no known competing financial interests or personal relationships that could have appeared to influence the work reported in this paper.

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
