# Peer review of "Numerical Study of Wheat Particle Flow Characteristics in a Horizontal Curved Pipe"

_processes, doi:10.3390/pr12050900_

Round 1

Reviewer 1 Report

Comments and Suggestions for Authors

The authors are suggested to clarify the following issues:

1) the model description in 3.1.4 is not for the DEM model.

2) The measurement of particle velocity must be detailed.

3) In addition to model validation, the comparison with measurements or the like should be made in the results and discussion, at least qualiatively.

Comments on the Quality of English Language

n/a

Reviewer 2 Report

Comments and Suggestions for Authors

1.    The agreement between the model and experiments shown in figure 10 is too good. How did the authors calibrate the DEM particles to achieve such good model predictions? The authors have used glued spheres to represent the DEM particles. So, there should have been a calibration routine to determine the CFD-DEM interaction parameters. The DEM particle calibration routine must be explained.

2.    The authors provide all the CFD and DEM equations. But they should also explain the information flow between the two models. What variables are being calculated by CFD and what variables are being calculated by DEM? And how are they exchanging information between the two? 

3.    Please provide how the time scale for CFD and DEM were determined? CFD has a larger time step compared to DEM. How did you fix the time steps for each? I did not see the CFD time step reported.

4.    Please provide details on how the CFD mesh size was determined compared to the DEM particle size. The authors say that the CFD mesh size should be larger than the particle size. What is the particle size? How did the authors arrive at the mesh size they used? 

5.    How much time did it take to simulate the model? And what kind of hardware was used?

Comments on the Quality of English Language

Minor editing of the language is advised. Also check for typing errors and fix them please.

Reviewer 3 Report

Comments and Suggestions for Authors

The paper titled ‘Numerical Study of Wheat Particle Flow Characteristics in a Horizontal Curved Pipe’ requires significant revision. The comments are given below:

(i)                  It is well known that in curved channels, gradual turns result in low pressure drops. Based on this, a geometry with R/D = 6 should have minimum pressure drop. However the authors suggest R/D = 2 to be optimal which is very much unexpected.

(ii)                Further, in figure 14, the contours Fig. (a)-(f) show that pressure drops are less than 100 Pa, however in Fig. 14g, the pressure is at least 10 times higher. For example for R/D = 1, the pressure at inlet from contours (Fig. 14a) is about 8 Pa whereas at outlet is -60 Pa (i.e pressure drop about 68 Pa). But in fig. 14g, the pressure drop is 2400 Pa. What is the reason for the discrepancy?

(iii)             As the particle mass flow is constant and diameter is constant, for continuous flow, the time-averaged particle velocity should be constant. Why particle velocity decreases with angle in Figs. 10, 12 and 13.

(iv)              Grid details (number of cells) and grid independent results are missing.

Comments on the Quality of English Language

NA

Round 2

Reviewer 1 Report

Comments and Suggestions for Authors

no further comments

Reviewer 3 Report

Comments and Suggestions for Authors

Fig. 10 shows plot of 'rate of speed loss'. Include relation/definition of this quantity in the paper

Comments on the Quality of English Language

Quality is fine
